# U.S. Physicians’ Training and Experience in Providing Trauma-Informed Care in Clinical Settings

**DOI:** 10.3390/ijerph21020232

**Published:** 2024-02-16

**Authors:** M. Lelinneth B. Novilla, Kaitlyn Tan Bird, Carl L. Hanson, AliceAnn Crandall, Ella Gaskin Cook, Oluwadamilola Obalana, Lexi Athena Brady, Hunter Frierichs

**Affiliations:** Department of Public Health, College of Life Sciences, Brigham Young University, Provo, UT 84604, USA; kt364@byu.edu (K.T.B.); carl_hanson@byu.edu (C.L.H.); ali_crandall@byu.edu (A.C.); egaskin@byu.edu (E.G.C.); damsel94@student.byu.edu (O.O.); bradyl99@student.byu.edu (L.A.B.); hunterthfrerichs@gmail.com (H.F.)

**Keywords:** trauma, traumatic experiences, adverse experiences, toxic/chronic stress, stress-related events, trauma-informed care, vulnerable populations, healthcare

## Abstract

Trauma-informed care (TIC) is a comprehensive approach that focuses on the whole individual. It acknowledges the experiences and symptoms of trauma and their impact on health. TIC prioritizes physical and emotional safety through a relationship of trust that supports patient choice and empowerment. It provides a safe and respectful healing environment that considers specific needs while promoting a greater sense of well-being, patient engagement, and partnership in the treatment process. Given the prevalence of trauma, this descriptive cross-sectional study examined the attitudes and perspectives of U.S. physicians (N = 179; 67% males; 84% White; 43% aged 56–65) in providing trauma-informed care using an anonymous 29-item online survey administered by Reaction Data. Findings showed that 16% (n = 18) of physicians estimated that >50% of their patients have a history of trauma. Commonly perceived barriers to providing TIC were resource/time/administrative constraints, provider stress, limited awareness of the right provider to refer patients who experienced trauma, and inadequate TIC emphasis in medical education/training. Expanding physicians’ knowledge base of trauma through training and organizational policy/support is crucial in enhancing their TIC competence, particularly in caring for patients with complex care needs whose social determinants increase their risk of exposure to adverse experiences that carry lasting physical and psychological effects.

## 1. Introduction

### 1.1. Epidemiology and Definition of Trauma

Trauma is pervasive. Six out of every ten men (60%) and five out of every ten women (50%) in the United States (U.S.) experience at least one form of trauma in their lifetime [1]. Out of every 100 Americans, 6 will experience post-traumatic stress disorder (PTSD) at some point in their lives [1]. Out of every 100 women, 8 (8%) experience PTSD compared to 4 out of every 100 men (4%) [1]. This heightened risk is partly attributed to the differences in traumatic events that women undergo, foremost of which is sexual assault [1].

Trauma can result from a variety of experiences. In a 2016 study, more than 70% of 68,894 respondents from 24 countries across six different continents reported having experienced a traumatic event, 30.5% of whom had exposure to 4 or more multiple traumatic events [2]. More than half of the reported traumatic exposures were a result of (1) witnessing death or serious injury, (2) untimely death of a loved one, (3) assault, (4) life-threatening automobile accident, and (5) serious illness or injury, with having a history of interpersonal violence as a strong predictor for experiencing future traumatic events [2]. 

Childhood exposure to trauma is far too common. Exposure to abuse, neglect, violence, abandonment, and other household challenges during the first 17 years of life, referred to as adverse childhood experiences (ACEs), can be traumatic for many [3]. Close to 64% of adults in the U.S. reported having experienced at least one type of adverse experience before 18 years of age. On the other hand, 17.3% or 1 in 6 adults, reported having four or more types of ACEs [3]. The social determinants that lead to health inequities also heighten the vulnerability to ACEs. Poverty and systemic racism, and the related limitations in educational and employment opportunities, are associated with ACEs [3]. For instance, ACES were highest among female Native Americans and Alaskans who were either unemployed or unable to work [3]. 

ACEs exert acute and long-term, intergenerational effects on health. Given the significant physical, social, and emotional changes that occur during the early stages of development, children and adolescents are susceptible to both the immediate and chronic effects of persistent toxic experiences. The negative impact on the developing brain can affect learning, attention, and decision-making. Children in dysfunctional households experience emotional challenges that include difficulties in establishing healthy and stable relationships, depression, or keeping a job [3]. In terms of young adult health and behavior, ACEs are predictive of anxiety, depression, and/or engagement in risky sex and substance use [4]. Such risky sexual behavior elevates the risk of teen pregnancy, pregnancy complications, fetal death, sexually transmitted infections, and sex trafficking [3]. The enduring and intergenerational negative impact of ACEs on physical and mental health occurs through the damage inflicted by the cardiovascular, immune, neuroendocrine, and metabolic sequelae of trauma on the body, ultimately resulting in multiple chronic conditions such as heart disease, cancer, diabetes, and suicide [3,5,6]. 

Chronic stress negatively affects the body. Allostatic load is the cumulative multi-system pathophysiological response of the body to chronic stress [5,7]. It explains the biological mechanisms through which persistent stress injures the body [5,6,7]. Research has repeatedly shown a graded relationship between the number of ACEs and the level of risk for various adverse health outcomes across the lifespan. The higher the number of ACEs, the stronger the association with sexual risk-taking, mental health problems, alcohol and drug use, and interpersonal and self-directed violence [8]. Recognizing the early clinical signs and symptoms of ACEs can reduce the continued exposure to trauma and the progression of serious health risks. 

### 1.2. Definition of Trauma

There are multiple definitions of trauma [9]. The Greek origin of the word trauma means “to wound” or “to pierce” [10,11,12]. Such wounds can be physical, as in physical trauma, or emotional, as in psychological trauma [13]. Thus, trauma can result from experiences that are physically damaging or emotionally overwhelming such as separation, divorce, loss/death, hunger, poverty, abuse, violence, elder abuse, rape, war/combat trauma, and ACEs [3,14,15].

Trauma is a highly individualized experience. As such, it triggers a response specific to the individual. The Substance Abuse and Mental Health Services (SAMHSA) introduces the concept of threat in its definition of trauma as the “physical, cognitive, and emotional response” from “an event, series of events, or set of circumstances that is experienced as…physically or emotionally harmful or life-threatening” [16] (p. 7) that can have a lifelong impact on one’s mental, physical, social, and emotional functioning, as well as spiritual well-being.

Trauma is the response to an overwhelming and disturbing experience. According to the American Psychological Association’s (APA) 2022 text revision of the fifth edition of its Diagnostic and Statistical Manual of Mental Disorders (DSM-5-TR), which describes and classifies mental disorders encountered in clinical, educational, and research settings, trauma is defined within the context of a post-traumatic stress disorder (PTSD) [17,18] and is described as the exposure to “death, threatened death, actual or threatened serious injury, or actual or threatened sexual violence”, resulting from a “direct exposure, witnessing the trauma, learning that the trauma happened to a close relative or close friend” or as an “indirect exposure to [the] aversive details of the trauma usually in the course of professional duties” [17]. Since DSM-5-TR states that “trauma- and stressor-related disorders” are distinct conditions resulting from exposure to a traumatic or stressful event that includes post-traumatic stress disorder (PTSD) [17,18], it establishes a clinical threshold for trauma with PTSD as the highest degree of severity [19].

Trauma is classified as to the duration of the exposure and the severity of the response that it elicits. Trauma could be a single incident or a complex one. Type I or single-incident trauma is a single-occurring event with a distinct start and ending, which allows the affected individual to seek help, move to a place of safety, and recover [11,20]. This includes accidents and natural disasters. Because such occurrence is usually publicly known, it leads to the validation and acceptance of the experience, which, in turn, reduces secrecy and shame [20]. On the other hand, a Type II or complex trauma results from prolonged, recurring, multiple, or cumulative events, in which there is limited time to recover between incidents [11,13,20,21]. This typically occurs in interpersonal relationships that are intended to be safe, usually perpetrated by an individual in a position of power, and which transpires in secrecy [20,21]. The power differential between the perpetrator and the individual, and the accompanying secrecy, can make it difficult and/or shameful to talk about the experience or to seek help, resulting in feelings of vulnerability and defenselessness [20,21]. A third type of trauma, Type III, was proposed by Heide and Solomon (1999) to further distinguish the type of complex trauma that starts early in life [22], during childhood or adolescence, which is ongoing, multiple, and pervasive, and is accompanied by injurious developmental sequelae [13,20,21].

Continuing and unresolved trauma carries significant and long-lasting cognitive, emotional, behavioral, and physical outcomes. Challenges in focus, memory, concentration, sleep, and mental health issues such as depression, anxiety, self-harm, and suicidal tendencies can occur [20,21]. It is likewise associated with chronic health conditions such as fatigue, body aches and pains, stomach and digestive issues, arthritis, Type 2 Diabetes, sexual difficulties, and unhealthy coping mechanisms, such as the use of drugs, tobacco, or alcohol, and/or eating disorders [20]. Because trauma can affect the individual’s sense of self, identity, and confidence, issues with trust, intimacy, and/or forming a secure attachment can occur, which can, in turn, hinder establishing a stable relationship [13,20].

There are various forms of trauma. SAMHSA’s *Trauma-Informed Care in Behavioral Health Services* (2014) distinguishes between naturally caused (disaster or cataclysmic natural events) and human-caused trauma (accidental or intentional acts) [23] (p. 35). Both categories are further subclassified into group or individual-level involvement. Naturally caused events include “tornadoes, lightning strikes, wildfires, avalanches, physical ailments or diseases, fallen trees, earthquakes, dust storms, volcanic eruptions, blizzards, hurricanes, cyclones, typhoons, meteorites, floods, tsunamis, epidemics, famines, and landslides or fallen boulders” [23] (p. 35). On the other hand, human-caused traumas that are accidental in nature include “train derailment, roofing fall, structural collapse, mountaineering accident, aircraft crash, car accident due to malfunction, mine collapse or fire, radiation leak, crane collapse, gas explosion, electrocution, machinery-related accident, oil spill, maritime accident, accidental gun shooting, and sports-related death, while intentional acts include arson, terrorism, sexual assault and abuse, homicides or suicides, mob violence or rioting, physical abuse and neglect, stabbing or shooting, warfare, domestic violence, poisoned water supply, human trafficking, school violence, torture, home invasion, bank robbery, genocide, and medical or food tampering, harassment, street violence, and bullying” [23] (p. 35).

Historical trauma and re-traumatization are other forms of trauma. Historical trauma is “the cumulative emotional and psychological wounding, as a result of group traumatic experiences, which is transmitted across generations within a community” [24]. In essence, historical trauma is a community-based experience, such as immense poverty, suffering, and loss, in which the residual impact affects generations of families and communities. Such trauma is typically associated with the experiences of ethnic or racial communities like American Indians and Alaska Natives [25], including immigrants and people of color [13].

Re-traumatization occurs when “clients experience something that makes them feel as though they are undergoing another trauma” [23] (p. 45). A traumatic stress reaction can be unintentionally triggered by treatment settings as well as by clinicians.

The understanding of trauma and its constructs is evolving. Finding consensus on a standard conceptual and/or operational definition of trauma has been challenging for academicians, researchers, and practitioners. The constructs, diagnostic criteria, boundaries, symptoms, forms of infliction, and the range and duration of the effects of trauma on the brain and on the body continue to change with the advances in behavioral science research and practice [13,26,27]. Some proponents argue that the existing definitions of trauma are either too restrictive and insufficient to capture its multi-faceted nature or are too broad and permissive that they conflate the boundaries between trauma and its correlates. As an example, the very term “trauma” has been used liberally within both categorical and dimensional constructs, each with its respective limitations [26,28]. Categorically, trauma is linked with PTSD based on the severity of the experience and/or the stressors [19]. As an example, DSM-5 TR’s definition of trauma is confined within the clinical threshold of PTSD, with PTSD serving as the uppermost boundary of severity [19]. Additionally, defining trauma based on its long-term negative impact on one’s self and psyche [28] conflates the clinical boundary between trauma and adversity [19]. This results in adversity being classified under trauma when, in reality, an adverse experience can result in heterogeneous pathological consequences, other than the trauma-specific psychopathology marked by vascular and cardiac hyper-reactivity [19,29]. Others argue that PTSD may not fully encompass the element of developmental trauma that is characteristic of adverse experiences [19,29]. Dimensionally, trauma can be seen from a more inclusive construct. It is defined as a stress response continuum in relation to an allostatic load while also incorporating the concept of post-traumatic growth (PTG) [26].

To reconcile both categorical and dimensional views, Krupnick [19] (p. 259) proposed a working definition of trauma founded on the stress theory, i.e., “to be considered traumatic, a stress response to an event must meet a necessary condition that the event be outside of the person’s normative life experience, and a sufficient condition that the response include a breakdown of self-regulatory functions” [19]. Krupnick considers two elements to be vital in defining trauma: (1) trauma as a “stress response” and (2) overwhelming effect of trauma on one’s coping ability, resulting in a sense of powerlessness or “loss of control over a situation” [19]. There is “breakdown of self-regulatory functions”, which in turn, disrupts the individual’s basic beliefs, expectations, and world views, which includes a sense of self, of “invulnerability”, of “benevolence”, and the belief of people being trustworthy [30]. The sense of helplessness characteristic of trauma was articulated by Freud. He described trauma as “the essence of helplessness that is brought about either externally or internally” [30,31] and which are “powerful enough to break through the protective shield” [31] (p.23).

Trauma can be re-experienced. Various reminders trigger “unwanted upsetting memories, nightmares, flashbacks, emotional distress, and physical reactivity” [12]. Trauma-related arousal can range from irritability/aggression, hypervigilance, and heightened startle reaction to difficulty concentrating, sleeping, and/or risky/destructive behavior [12]. Negative thoughts and feelings are manifested in the “inability to recall key features of the trauma, having overly negative thoughts and assumptions about oneself or the world, exaggerated blame of self or others for causing the trauma, negative affect, decreased interest in activities, feeling isolated, [and the] difficulty [of] experiencing positive affect” [12].

Acute stress disorder and PTSD are trauma-related disorders. These conditions share the same clinical presentation but differ only in the duration of symptoms. Acute stress disorder is characterized by the immediate onset of symptoms that lasts from 3 days to 4 weeks following a traumatic event [32]. On the other hand, PTSD lasts for more than a month and causes functional impairment [12,13,14], such as the relentless re-experiencing of the event through repeated recollections, negative thoughts or feelings, increased arousal, and/or avoidance of trauma-related stimuli that serve as reminders of the event.

PTSD can occur at any age [33]. The National Center for PTSD estimates that approximately 6 out of 100 individuals can develop PTSD at some point in their lives [1,33]. The use of validated measures for adults and children, such as those created by the National Center for PTSD, is required to assess trauma exposure [1].

Both biological and extraneous factors can increase the risk for PTSD. Certain genetic polymorphisms and epigenetics (such as the inability to discriminate and extinguish fear memories) [34], age (childhood exposure), gender (females more than males), type of trauma (physical and sexual assault), intensity (presence of stress, co-existing depression, and psychosis), and chronicity of the traumatic event can increase the likelihood of experiencing PTSD [1,33]. However, not all individuals who experience a traumatic event develop PTSD. Certain social and personal characteristics are less vulnerable to severe post-traumatic symptoms, which makes PTSD less likely [1,33,35]. These protective factors include social support [1,33,35,36,37], adaptive coping [35,37], optimism [35,38], trait resilience [34,35,39], self-efficacy [35,40], and secure attachment [35,41]. Based on meta-analyses and systematic reviews, these factors can help individuals “bounce back” [42] and undergo positive post-traumatic growth (PTG) that is marked by a higher level of psychological functioning [35,43].

### 1.3. TIC Pyramid and the Principles of a Trauma-Informed Approach to Care

Trauma is defined by three “E’s”: the event, the experience, and the effect [16]. The event is the actual incident, either a single or repeated occurrence, which presents a physical or psychological threat or harm with acute or persistent adverse physical, social, emotional, or spiritual effects on the individual’s functioning [16]. How the event is experienced and interpreted by the individual determines whether it is a traumatic occurrence, particularly within the context of a power differential. The experience may elicit feelings of self-blame, shame, guilt, humiliation, betrayal, fear, or of being silenced. The effects of such experience may range from the inability to trust in relationships, cope with daily stresses, and regulate behavior and emotions to hypervigilance, numbing, or avoidance [9]. The three-“E”-framework serves as a guide for healthcare providers, including psychologists and mental health counselors, in identifying individuals with traumatic experiences.

Trauma-informed care (TIC) is a holistic approach. It acknowledges the experience of trauma, its signs and symptoms, and its impact on health and functioning while recognizing the need for individualized treatment, support services, and resources [16,25,44,45]. The TIC pyramid serves as a framework for applying trauma-sensitive practices in clinical settings. It is based on a literature review of peer-reviewed journals on trauma care. The five tiers of the pyramid correspond to the five core TIC principles in healthcare: (1) “Patient-Centered Communication and Care”, (2) “Understanding the Health Effects of Trauma”, (3) “Interprofessional Collaboration”, (4) “Understanding Your Own History and Reactions”, and (5) “Screening. In turn, each core TIC principle is comprised of recommended strategies for clinical practice. The five core principles are further classified under one of the two major TIC domains: (1) “Universal Trauma Precautions” and (2) “Trauma-Specific Care.” The “Universal Trauma Precautions” domain is comprised of the first two principles of “Patient-Centered Communication and Care” (base of the pyramid) and of “Understanding the Health Effects of Trauma” (second tier) [15]. On the other hand, the “Trauma-Specific Care” domain starts from the third to the fifth tier or apex of the pyramid and highlights the principles of “Interprofessional Collaboration”, followed by “Understanding [Personal Trauma] History and Reactions”, and finally, by “Screening” [15]. See Figure 1.

Trauma-informed care is grounded on six key principles. These include (1) safety, (2) trustworthiness and transparency, (3) peer support, (4) collaboration and mutuality, (5) empowerment, voice, and choice, and (6) cultural, historical, and gender issues [16]. The first principle of TIC practice is safety. Maintaining physical and psychological safety through trust and transparency is crucial in TIC-based patient interactions [16]. Peers, such as other trauma survivors, provide mutual support [16]. Moreover, organizations employing TIC provide gender-responsive services that incorporate policies, protocols, and processes that disapprove of stereotypes and cultural biases [16].

Trauma-specific interventions promote healing by promoting patient voice and choice. Partnership and shared decision-making are emphasized to equalize the power difference between the patient and the care team. TIC also fosters healing through resilience and prevention of re-traumatization by recognizing and avoiding or altering procedures and examinations that can trigger memories of the adverse experience. It applies trauma-specific interventions, such as assessments, treatments, and recovery supports, while also incorporating the principles of trauma-informed care into the organizational culture and policies [16].

Clinical trauma-informed care is tailored to the needs of the patient. Routine screening for trauma history identifies patients who can benefit from individualized trauma-sensitive support services and resources [15]. In contrast to the traditional “trauma-blind” delivery of services, TIC incorporates the four R’s: (1) realizing the impact of trauma and helping the individual recover, (2) recognizing the signs and symptoms of trauma, (3) responding by utilizing TIC principles in all areas of services, and (4) resisting re-traumatization [16,46]. Thus, in a trauma-informed approach, the provider asks, “What happened to you?” instead of “What’s wrong with you?” [47]. Instead of disregarding the hurt or downplaying its intensity, asking, “What’s wrong with you?” helps reassure the patient that he/she is not at fault for having undergone a harrowing experience [48].

### 1.4. Trauma-Informed Care in Complex Care Patients

Exposure to trauma disproportionately impacts patients with complex care needs. Complex care patients are individuals whose multiple chronic physical and behavioral conditions are made worse by social inequities, such as food, housing, transportation, and/or job insecurity, unemployment, poverty, homelessness, racism, ageism, and others [49]. Many come from vulnerable or marginalized segments of society: older adults, individuals and/or households with incomes below the federal poverty level; individuals and families from indigenous or multicultural backgrounds; persons with disabilities; medically underserved individuals with substance use disorder; the homeless, those on parole, detained, or incarcerated; individuals who identify as LGBTQ [49].

#### 1.4.1. Trauma-Informed Care among Indigenous People

TIC recognizes both individual and collective trauma. Substance use among indigenous populations can be traced not only to unresolved personal grief but also to group intergenerational trauma. Treatment based on conventional medicine can undermine the benefits of cultural healing practices [50], while trauma-sensitive care combines allopathic with cultural healing practices to manage the root causes of substance use disorders for better patient outcomes [51,52].

#### 1.4.2. Trauma-Informed Care in Pregnancy and Childbirth

Patients with a history of trauma and ACEs are more likely to have complex care needs later in life. ACEs can heighten the risk for PTSD during pregnancy, occurring almost five times higher in low-resource settings [53]. For some women with ACEs, re-traumatization can occur as the birthing experience may trigger past experiences of abuse. Instead of an anticipated event, childbirth may be perceived as intrusive, disturbing, or even overwhelming. This can lead to PTSD during the postpartum period, which may interfere with childrearing and mother–child bonding while increasing the risk for future premature and/or low-birth-weight pregnancies [53]. Routine TIC screening in this setting provides a safe space that can increase patient disclosure of sexual abuse or intimate partner violence by almost 60% [54]. Additionally, nurse home visits, combined with psychotherapy, emotional regulation, mindfulness, and self-compassion, can reduce the negative impact of trauma on both the mother and the child [53].

#### 1.4.3. Trauma-Informed Care among Military Veterans

Military veterans have an elevated risk for PTSD. The Veterans Health Association incorporates TIC approaches during telemedicine and tele-mental health consults, particularly when access to care was limited by the COVID-19 pandemic [55]. These virtual TIC consults have been found to be as effective in providing patient safety and transparency as the in-person visits [55].

### 1.5. Study Aims

Gaps exist in integrating TIC into routine clinical care. Despite the pervasive nature of traumatic experiences, TIC has yet to be widely integrated into clinical settings. In this study, we examined the attitudes and perspectives of U.S physicians in providing trauma-informed care by determining the following: (1) possible indicators of trauma history in a patient; (2) estimated patient caseload with a history of trauma; (3) perceived barriers to implementing TIC in clinical care; (4) extent of physician training in the five core TIC principles; (5) extent of application of various TIC strategies based on Raja et al.’s TIC pyramid [11]. Given the prevalence of adverse experiences in the U.S. population, the findings of this study may likewise shed light on the existing gaps in trauma-informed education, practice, research, and policy in healthcare settings.

## 2. Materials and Methods

### 2.1. Study Design

We conducted a descriptive cross-sectional study using a national sample of U.S. physicians who completed an anonymous 29-item online survey to determine their perspectives, training, and experiences in providing trauma-informed care.

### 2.2. Study Subjects and Recruitment

The anonymous 29-item online survey was administered by Reaction Data, a Utah-based healthcare market research firm. Following approval from the Brigham Young University (BYU) Institutional Review Board (IRB#2022-373) on 27 September 2022, Reaction Data administered the survey from 22 November 2022 until 9 February 2023. The informed consent, added to the front page of the survey, was read and signed by respondents before participation.

The respondents in this study were drawn from Reaction Data’s database, which is comprised of various healthcare providers (MDs, DOs, NPs, PAs) and healthcare workers (nurses and other healthcare professionals) in the United States who were accustomed to receiving and completing pulse surveys from Reaction Data. This database lists 13,532 U.S. physicians, which served as the sampling frame for this study. These physicians were working in U.S. healthcare facilities such as hospitals, private practices, community clinics, outpatient centers, surgical centers, urgent care centers, long-term care facilities, rehabilitation centers, and academic medical centers. The research team created three email invitations, each of which contained brief information about the study and the survey link, which were then sent out sequentially by Reaction Data to all physicians who were the target respondents in this study. The first email invite was distributed on 22 November 2022. Once the link was clicked and the survey was completed, no additional invites were sent out to these respondents. The second and third email invites were sent out about a month to a couple of months, respectively, from the first invitation to those who had not yet opened the link nor completed the survey. No incentives were used to encourage participation.

### 2.3. Survey Instrument, Validity Testing, and Administration

The survey instrument, which was subsequently referred to by the authors as the Trauma-Informed Care Physician Assessment Tool (TIC-PAT), was developed by the research team (KTB, CLH) with guidance from Prof. Sheela Raja, an expert in the field of trauma-informed care. The survey was comprised of four demographic multiple-choice questions on gender, age, race, and ethnicity and 22 5-item Likert scale questions with options ranging from “Always”, “Often”, “Sometimes”, “Rarely”, and “Never” or “Strongly Agree”, “Agree”, “Neither Agree nor Disagree”, “Disagree”, and “Strongly Disagree”. These 22 questions were based on the five core TIC principles and their corresponding strategies from Raja et al.’s [15] TIC Pyramid in healthcare and were broken down in the survey as follows: (1) Patient-Centered Communication and Care (6 questions); (2) Understanding the Health Effects of Trauma (3 questions); Interprofessional Collaboration (4 questions); Understanding Your Own History and Reactions (4 questions); and Screening (5 questions). Three other questions were added on the estimated percentage of patient caseload with traumatic history (multiple choice), healthcare provider’s training on trauma-informed care (yes, no, unsure), and factors that inhibit the practice of trauma-informed care in clinical settings (multiple response questions using a 5-point Likert scale) (see Appendix A).

### 2.4. Data Analysis

The survey data were temporarily stored by Reaction Data and subsequently sent to the BYU research team for analysis as a CSV download. The data were analyzed using SAS 9.4. The frequencies, percentages, mean scores, and standard deviations were calculated for each of the five TIC Pyramid principles and their respective strategies.

## 3. Results

A total of 179 U.S. physicians (1.3% response rate) responded to an anonymous 29-item online survey on trauma-informed care, with 80% (143/179) of respondents completing each question.

### 3.1. Demographic Characteristics

All 179 respondents were physicians, 75 (67.0%) of whom were males and 35 (31.3%) were females, with 1 respondent (0.9%) who identified as a transgender female. In terms of age, the majority (48; 42.9%) were 56–65 years old, followed by those who were either 46–55 (25; 22.3%) or 66–75 years old (24; 21.4%). Most respondents (82; 83.7%) identified as White or Caucasian, while the rest identified as Asian or Pacific Islander (9; 9.2%), Multiracial or Biracial (4; 4.1%), Black or African American (2; 2.0%), or Middle Eastern (1; 1.0%). In terms of their employment position, specialty physicians comprised a little more than half of the respondents (92; 51.4%), and they represented the following specialties: surgery and anesthesiology, critical care medicine, reproductive medicine, ophthalmology, otolaryngology, psychiatry/neurology, cardiology, dermatology, gastroenterology, hematology, oncology, and nephrology/urology. On the other hand, primary care physicians coming from general internal medicine, pediatrics, obstetrics/gynecology, and complementary medicine (chiropractic care) made up less than half of the respondents (80; 44.7%). The rest identified as either a medical director or department chair (7; 3.9%) (see Table 1). The demographic characteristics of respondents in this study aligned with the demographic characteristics of U.S. physicians for gender, where 61% are male, most are white physicians (66%), and the largest distribution of physicians is between the ages of 56 and 66 [56].

### 3.2. Attitudes and Perspectives of U.S. Physicians in Providing Trauma-Informed Care

#### 3.2.1. Experiences and Training in Trauma-Informed Care


**Estimated Percentage of Patient Caseload with a History of Trauma**


Study participants were asked to estimate the percentage of their patients who experienced a traumatic event, such as sexual trauma, interpersonal violence, community violence, and other adverse circumstances. Only 16% of physicians (18/111) reported that more than 50% of their patients had a history of trauma, while 42% (47/111) claimed that fewer than 30% of their patients experienced trauma, and 39% (43/111) believed that the prevalence would be within 30% to 50% of their patient caseload.


**Perceived Barriers to Implementing TIC in Clinical Settings**


Physicians were asked about the factors that inhibited their ability to practice TIC. This was asked using a dropdown question that displayed a list of nine multiple-choice answers. The top five factors that physicians “Strongly Agreed” or “Agreed” with as barriers to TIC implementation in clinical settings included the following: (1) resource constraints such as limited in-network services, long wait lists, shortage of culturally and linguistically appropriate services, or lack of geographically accessible services and resources (62/117; 53%); (2) referral limitations due to the lack of familiarity of the right provider to refer patients who have a history of trauma; (3) time constraints such as scheduling conflicts, short appointment time, and a high patient load (55/118; 46.6%); (4) administrative constraints such as the expectation to see a high patient load or complete paperwork (49/117; 41.9%); (5) the difficulty of managing personal stress and/or practicing self-care strategies (45/117; 38.5%). This was closely followed by inadequate emphasis on TIC during medical education/training (45/118; 38.1%) (see Table 2).


**Extent of Physician Training in the Five Core TIC Principles**


All physicians who answered this question reported being trained in at least one TIC core principle (see Table 3). Based on the results, almost a quarter of the respondents received training on either three (27/112; 24.1%) or all five core TIC principles (26/112; 23.2%).

Although 23% of physicians reported being trained in all five core TIC principles, the most frequently received training, by far, was on universal trauma precautions, which advocate for both patient-centered communication and care and the understanding of the health effects of trauma. This was followed by training in trauma-specific care practices such as screening and interprofessional collaboration. On the other, physicians reported the least amount of training on understanding how their personal history of trauma could affect them on a personal level and their own reactions to a patient’s trauma. This principle also espouses the exercise of self-care strategies, such as seeking counseling to manage high levels of stress (see Table 4).

#### 3.2.2. Extent of Application of Various TIC Strategies: Practicing TIC Patient-Centered Communication and Care

*Summary of Results*. To determine the extent in which physicians practiced “Universal Trauma Precautions” through patient-centered communication, the following six questions were asked in the survey: (1) inquiring about what can be done for a patient to feel more comfortable during the appointment; (2) prior to a physical examination, presenting a brief summary of the parts of the body that will be examined and allowing the patient to ask questions; (3) giving the option of shifting an item of clothing out of the way rather than having a patient put on a gown when an entire area does not need to be examined; (4) offering a patient a pillow for her/his back if they are anxious about being in the supine position; (5) giving a patient the option of a mirror to see procedures or examinations that are out of her/his field of vision; finally, (6) offering a patient a way to verbally and non-verbally “signal” anxiety (e.g., raising hand) during procedures/exams (e.g., Pap smear).

The majority reported “Always” carrying out five of the six strategies under “Patient-Centered Communication.” However, almost half (49%) of the physicians admitted to “Never” having provided a mirror for patients to visualize the procedures or examinations outside their field of vision. Strategies that physicians reported as carrying out either “Always” or “Often” included the following: (1) asking what can be done to make the patient comfortable during the appointment; (2) presenting a brief summary of the parts of the body that will be examined and allowing the patient to ask questions; (3) giving the option of shifting an item of clothing out of the way rather than having the patient put on a gown when the entire body does not have to be examined; (4) offering the patient a pillow for her/his back if anxious of being in a supine position; finally, (5) offering the patient a way to “signal” anxiety verbally or nonverbally (see Table 5).

#### 3.2.3. Extent of Application of Various TIC Strategies: Understanding the Health Effects of Trauma

*Summary of Results.* The extent to which physicians recognized the health effects of trauma was assessed through three questions regarding the following: (1) identifying the possible indicators of trauma history in a patient (multiple response questions with the following options: anxiety, insomnia, difficulty trusting others, high intake of sugary food and drink, smoking, drinking, substance abuse (illicit or prescription), engaging in unprotected sex, overeating, or none of the above); (2) discussing with the patient the relationship between unhealthy behaviors (maladaptive coping methods) and stress/trauma; (3) brainstorming with the patient potential solutions for changing behavior detrimental to well-being.

There was a general recognition among physicians of the various indicators of a history of trauma in a patient. Out of 160 respondents, 47 (29.4%) physicians identified all of the nine listed conditions as indicators of traumatic experiences, while 15 (9.4%) chose eight out of the nine options by leaving out the high intake of sugary food and drink. On the two remaining questions under this particular TIC principle, 90 out of 159 physicians (57%) responded as “Sometimes”, “Rarely”, or “Never”, discussing with their patients the relationship between unhealthy behaviors (maladaptive coping methods) and stress/trauma. Additionally, half of the physicians (79; 50%) reported “Sometimes”, “Rarely”, or “Never” brainstorming with their patients regarding potential solutions for changing behavior detrimental to their well-being (see Table 6).

#### 3.2.4. Extent of Application of Various TIC Strategies: Collaborating Interprofessionally in Practicing Trauma-Informed Care

*Summary of Results.* The extent to which physicians collaborated interprofessionally when practicing trauma-informed care was assessed through four questions that asked about the following: (1) maintaining a list of referral sources for patients who disclosed a trauma history; (2) confidence in sensitively referring a patient with trauma; (3) availability in the waiting room of referral and educational materials about trauma; finally, (4) having the confidence in working with nurses, medical interpreters, first responders, and others when caring for patients who have experienced trauma.

More than half of the respondents “Strongly Agree[d]” or “Agree[d]” to the following: (1) maintaining a list of referral sources for patients who disclosed a trauma history (84; 56%); (2) being confident in referring a patient with trauma (90; 59.6%); (3) working with nurses, medical interpreters, first responders, and others when caring for patients who have experienced trauma (84; 55.3%). However, more than half of the respondents admitted to “Disagree[ing]” or “Strongly Disagree[ing]” to having readily available referral and educational materials about trauma in the patient waiting room (78; 52.0%) (see Table 7).

#### 3.2.5. Extent of Application of Various TIC Strategies: Understanding One’s Personal Trauma History and Reactions

*Summary of Results*. The physicians’ level of understanding of their personal trauma history and the response was determined through the following four questions: (1) reflecting on [personal] stress and/or trauma history and how it may influence interactions with patients; (2) recognizing when caring for patients with a trauma history begins to impact personal emotional health and wellbeing; (3) practicing self-care strategies (e.g., exercise, social support, etc.); finally, (4) using counseling/mental health services to help manage stress when experiencing high levels of stress.

Among all the core TIC principles, this four-item principle obtained the lowest internal consistency or reliability with a Cronbach’s alpha score of 0.51. Overall, physicians “Strongly Agree[d]” and “Agree[d]” on thinking about their personal stress/trauma and realizing when it began to impact their own emotional health and well-being. However, when it came to actually managing personal stress by seeking professional counseling or mental health services, particularly when tension was high, more than half (58%; 69/119) of physician respondents in this study admitted to “Rarely” or “Never” having sought such services (see Table 8).

#### 3.2.6. Extent of Application of Various TIC Strategies: Screening for Trauma in New Patients and in Patients with Functional Challenges

*Summary of Results*. A history of trauma in new patients and in those with disabilities may go undiagnosed. The extent to which physicians consistently screened for a history of trauma in new patients and in those with functional difficulties was determined through five questions regarding the following: (1) screening for trauma in every new patient; (2) the importance of assessing every case of trauma; (3) screening for current trauma or history of trauma following a universal screening; (4) preparing patients for potentially difficult questions prior to a trauma screening (e.g., reviewing confidentiality); (5) providing all office staff the communication skills training on how to sensitively talk to patients who disclosed a history of trauma.

There was an overwhelming agreement on the importance of assessing every case of trauma. Of the 142 physicians who responded to this question, 116 (81.7%) “Strongly Agree[d]” or “Agree[d]” to this statement. On the other hand, responses were almost evenly distributed along the spectrum of “Always”, “Often”, or “Sometimes” in terms of (1) screening for trauma in every new patient, (2) screening for current or past history of trauma after a universal screening, (3) preparing patients for potentially difficult questions prior to a trauma screening, and (4) providing training to all office staff on how to sensitively talk to patients who disclosed a trauma experience (see Table 9).

## 4. Discussion

Trauma leaves lasting emotional and psychological scars. The gravity of such experience can impact both mind and body and shape health outcomes and overall quality of life. A trauma-informed approach focuses on the whole person and considers how a history of trauma can affect a patient’s physical, emotional, and mental health and well-being, as well as the ability of healthcare providers to offer trauma-informed care.

### 4.1. Clinicians’ Perspectives on Implementing Trauma-Informed Care in Clinical Settings

Basic to trauma-informed care is recognizing how trauma affects the patient. Rather than following a traditional treatment protocol, a trauma-informed approach serves as a comprehensive framework of interventions [46] directed at multiple levels: medical education and training, professional practice, and institutional/organizational levels to address the prevalence and impact of trauma on both patients and healthcare providers.

In this descriptive study on a national sample of U.S. physicians, we used Raja et al.’s [15] TIC pyramid to examine the extent to which trauma-informed care practices were applied by U.S. physicians in clinical settings.

The TIC pyramid serves as a translational framework for applying trauma-sensitive care strategies in healthcare [15]. It has been used by other researchers as a benchmark for assessing the adequacy of a TIC-based curriculum in primary care [57]. In addition to recommending clinical care strategies, the TIC pyramid lists potential research questions, study designs, and primary outcomes for addressing the gaps in TIC research.

The TIC pyramid has two overarching clinical implementation domains [15]. The first domain, “Universal Trauma Precautions”, is comprised of general clinical and health system practices applied to all patients, regardless of the provider’s knowledge of the patient’s trauma history [15]. These are intended to foster trust and rapport by reducing fear or anxiety. For instance, even among patients without a history of trauma, fear or discomfort may be experienced when undergoing medical procedures. For those who have experienced trauma, these very same procedures can elicit a greater level of anxiety, particularly when being alone with a healthcare provider and/or when their bodies are seen, touched, and analyzed as part of the clinical examination process.

The second domain, “Trauma-Specific Care”, lists techniques tailored for patients with a history of trauma that include interprofessional collaboration, conducting universal screening for traumatic events, provider awareness of vicarious victimization, professional burnout issues, and an understanding of how one’s trauma history can affect a provider’s interactions with patients [15]. The latter is particularly crucial as healthcare providers are at risk for secondary traumatic stress when interacting with patients who have experienced trauma [9,58,59,60]. In physician–patient interactions, the healthcare provider acts as a “trauma-steward” [15], providing the best patient care while also exercising self-care to avoid being drawn into the vortex of the patient’s struggles. Secondary traumatic stress, also referred to as “compassion fatigue” or vicarious trauma, is “emotional duress that results when an individual hears about the firsthand trauma experiences of another” [61]. This can lead to difficulties in providing quality care, as well as in staff turnover, due to chronic fatigue, burnout, poor concentration, disturbing thoughts, avoidance, absenteeism, and physical illness [9]. Studies have shown that the risk factors for secondary traumatic stress among medical personnel include low job satisfaction, regret [60], low resilience, history of personal trauma, being female, and spending 10% or more of one’s time with trauma patients [59].

The overall findings of this study point toward limited physician training and experience in trauma-informed care in clinical settings. Although respondents claimed to have received training on three to five core TIC principles in healthcare, such claim of training did not always correspond with the reported application of trauma-sensitive care strategies in clinical settings. Notwithstanding, this study was based on Raja et. al.’s [15] TIC pyramid. Although the TIC principles in healthcare are relatively generic across existing clinical TIC curricula, there may be nuances in TIC principles and practices that may not have been explicitly captured by Raja et al.’s TIC pyramid [15]. It is also possible that during medical education and/or professional training, some TIC concepts and practices may have been largely emphasized while some may not have even been discussed at all.

There was a low estimate of trauma history in patient caseloads in this study. When physicians were asked to estimate the percentage of their patient caseload who have experienced trauma, only 16% of respondents reported that more than 50% of their patients experienced trauma despite the high prevalence of trauma in the U.S. population. This could reflect either the lack of familiarity with how pervasive trauma was among their patients and/or the lack of routine screening for trauma history in patients. Nonetheless, it is important to note that the physician respondents in this study were not asked to use a screening tool or diagnostic instrument to assess the prevalence of trauma in their patients. Although ideal, this would entail conducting a separate study as the survey length alone already presented a time constraint for some respondents. Instead, the physicians in this study were asked, through a survey question, to estimate the percentage of their patients with trauma history. As such, their responses were subject to recall bias, leading to either an under- or over-estimation of the actual value.

On “Patient-Centered Communication and Care”, subjects reported either “Always” or “Often” in implementing almost all six recommended TIC practices except in offering a mirror for patients to see procedures or examinations outside their field of vision. This finding may reflect the fact that these general trauma strategies, which are classified under the “Universal Trauma Precautions”, are typically incorporated and widely supported as part of the standard of care, while more specific trauma interventions have yet to be fully integrated into standard clinical guidelines. Compared to universal trauma precautions, highly specific trauma strategies require follow-up training and concerted multi-level adoption and endorsement. Nonetheless, even within the set of strategies for patient-centered communication and care, it was revealed that certain practices were not commonly applied, such as offering a mirror to patients for them to visualize the execution of a medical procedure that is beyond their range of vision.

Since it is possible that TIC training and implementation may vary by specialty, a separate study could examine the relationship between the level of TIC training and implementation in medical specialties and subspecialties. Seeing that primary care physicians, such as those in family medicine, internal medicine, pediatrics, geriatrics, and obstetrics and gynecology, serve as gatekeepers to more specialized care and are more likely to see a diverse group of patients, they may be more open to adopting a trauma-informed care approach. Conversely, subspecialists, such as those in surgery and anesthesiology, typically receive patients via referrals from their primary care physicians. The nature of their services and interactions with patients are far more specific, although these do not necessarily preclude the practice of trauma-informed care.

Another finding worth noting pertains to the principle of “Interprofessional Collaboration.” In this study, one of the most commonly identified barriers to implementing TIC in clinical settings was referral limitations due to the lack of knowledge of the right provider(s) to whom physicians can refer their patients with a history of trauma. Yet, this perceived barrier contradicted the responses on implementing the elements of “Interprofessional Collaboration”, in which most physicians answered “Always” or “Often” when asked about (1) maintaining a list of referral sources for patients who disclosed a trauma history, (2) confidence in sensitively referring a patient with trauma, and the (3) confidence in working with an interprofessional team in caring for patients who have experienced trauma. It is possible that although physicians are aware of trauma services and resources available in the community, they may not be exactly knowledgeable of the specific providers and referral agencies with the expertise in providing evidence-based trauma services [16]. Regarding interprofessional referrals, SAMHSA recommends establishing a system of cross-sector communication through inter-agency collaboration and training on trauma-informed approaches [16]. Challenges in referral issues may also be addressed by looking into evidence-based practices of other healthcare facilities, agencies, and programs. For instance, the Women’s HIV Program at the University of California, San Francisco, implements various initiatives that incorporate the patient voice in addressing trauma and offers an on-site trauma staff with a functioning referral system to trusted partners [62]. Moreover, Menschner and Maul [9] advised partnering with patients and their families in planning the treatment process and in inviting their feedback on trauma-informed organizational policies and work culture.

A history of trauma may go undiagnosed. On the TIC principle of “Screening”, there was a general agreement among physician respondents on the importance of assessing every case of trauma. However, instead of an overwhelming affirmative reaction to the strategies for screening new patients and those with functional challenges for current or past trauma, the responses were evenly distributed as to “Always”, “Often”, or “Sometimes”. This may reflect the need for both basic and follow-up training [9]. For instance, in “Understanding the Health Effects of Trauma”, the findings potentially reflect the need for promoting physician-patient discussions on identifying unhealthy behavior, recognizing the link between trauma and maladaptive coping behavior, and exploring resilience-building strategies, interventions, and programs.

### 4.2. Trauma-Informed Care as an Under-Emphasized and Under-Implemented Approach in Healthcare

Several factors can promote as well as hinder TIC implementation in clinical settings. The results of this study point to resource, time, administrative, and patient referral limitations as the most common barriers to TIC implementation. These perceived barriers align with the findings of the systematic review by Huo et al. [63] on the factors that impede TIC application in healthcare settings. For instance, the perceived and actual relevance of TIC in the health setting can be shaped by both internal and external influences: staff and leadership buy-in, financial and staffing resources, organizational policies and procedures, institutional engagement, and interagency partnerships [63]. Other factors identified in this systematic review pointed to individual-level elements, such as provider readiness versus resistance to change, and implementation-level elements that included organizational flexibility versus rigidity to change, accessibility of training, conducting data review and program evaluation, and assessing the overall alignment of TIC initiatives with organizational policies [63]. These factors not only serve as key program targets for implementing TIC in clinical settings, but also as benchmarks for evaluating the impact of TIC on patient health outcomes. Identified barriers and promoters of TIC practice can guide healthcare facilities in developing interventions tailored to the needs of the patients and the communities they serve. There is a disparity between physicians’ TIC education and the reported extent of their TIC training. Based on the findings of this study, there is a discrepancy in the survey responses between the actual TIC preparation received in medical school and professional practice and the physicians’ reported extent of TIC training. Although physicians in this study reported being trained in three to five core TIC principles of the TIC Pyramid, they also identified an insufficient emphasis on TIC in medical school and/or professional training, leading to an impediment in TIC practice in clinical settings. To offset such inadequacy, medical school curricula could be reviewed and updated to expand physicians’ knowledge base of trauma and secondary trauma and of various upstream approaches to trauma prevention.

At the practice level, training models have been developed to incorporate TIC in healthcare. Similar to Raja et al.’s conceptual framework [15], Roberts et al. [64] created a TIC pyramid tailored to primary care. This trauma-informed primary care (TIPC) model emphasizes patient empowerment and empathy as the foundation of a trusting relationship between the provider and the patient while also allowing the patient personal control over her/his healthcare choices. Although TIC curricular training frequently covers the health effects of trauma, other TIC principles are either underemphasized or undiscussed due to scheduling conflicts and the variability in the level of TIC experience among physicians receiving the training (i.e., residents vs. practicing physicians) [57].

In 2021, the American Medical Association (AMA) adopted a policy that supports a trauma-informed care approach in all medical settings [65]. In conjunction with this policy, the AMA recognizes the symptomatology of trauma, its negative effects on patients, and the value of integrating trauma knowledge into policies, procedures, and practices in treating patients and preventing re-traumatization [65]. The AMA supports evidence-based prevention strategies and screening for ACEs, given its link with suicide, substance use disorders, and various chronic conditions. In addition, the AMA supports incorporating TIC in both undergraduate and graduate medical curricula, expanding research on ACEs, and translating research findings into clinical and public health interventions [65]. A valuable advice that the AMA dispenses for physicians is not only about being educated on TIC but also being “train[ed] [on] what not to do” in responding to trauma, such as forcing a disclosure [66]. In addition, the AMA explains that TIC allows physicians to observe “feelings of shame, fear or avoidance” in their patients and uncover their underlying factors while also utilizing such information to guide treatment [66]. Ultimately, the AMA’s major aim is to universally apply TIC principles to all patients. Nevertheless, despite the AMA’s adoption of a policy recognizing TIC in all medical settings, the universal adoption and implementation of TIC remain to be seen. Moreover, the policy does not articulate how physicians’ own trauma history can affect their delivery of TIC, nor does the policy stipulate system- and organizational-level resources that physicians with a significant trauma history can avail.

Gaps remain in increasing physicians’ recognition of the interplay between their personal trauma and their reaction to the trauma disclosed by their patients. In this study, training was reported to be the lowest for this particular core TIC principle on “Understanding Own [Personal Trauma] History and Reactions.” This is a crucial finding, given the overlap between compassion fatigue and physician burnout. Both compassion fatigue and physician burnout are considered occupational hazards [67,68]. These reflect the psychological and emotional toll that providing care takes on physicians. These are the hidden costs of caregiving that can impact personal well-being, quality of patient care, treatment decisions, and job satisfaction, including the overall delivery of care [67,68]. Those dealing with compassion fatigue tend to offer themselves fully to their patients and their work to the detriment of their own personal well-being [67] while those experiencing burnout become more emotionally withdrawn [67,68]. In both cases, emotions affect the practice of medicine and it can become challenging to balance objectivity and empathy. Over time, unaddressed compassion fatigue and physician burnout can lead to emotional detachment [68]. In terms of internal consistency, the recommended TIC practices under this principle were the least related as a group, with a Cronbach’s alpha of 0.51. This may be due to the fact that this principle attempted to examine the various facets or dimensions of a provider’s own history of trauma such as: (1) a provider’s understanding of her/his personal trauma, (2) impact of personal trauma on a provider’s emotional health, (3) impact of personal trauma on the quality of a provider’s interactions with patients, and (4) a provider’s response to stress using self-care strategies and professional counseling.

### 4.3. Implications of the Findings on TIC Implementation in Clinical Settings

Based on the findings of this study, advancing TIC in clinical settings may need to consider the following key points:(1)Enhancing physician TIC competence on all five core TIC principles through interventions directed at multiple levels of medical education and clinical training.

Medical education curricula could be updated to incorporate TIC. This will deepen the understanding of future and current providers on the acute and long-term effects of trauma across the lifespan [69,70]. At the practice level, peer TIC mentoring, modeling, and support can increase the adoption and implementation of TIC in clinical settings. At the institutional level, TIC could be integrated throughout healthcare by prioritizing and allocating resources for TIC training, with incentives granted to physicians adopting such approach in their practice [9,71,72]. Institutional policies could be reviewed and updated regularly to increase organizational-level recognition of the effects of trauma on both patients and providers while also addressing the challenges of integrating trauma-informed care in clinical practice [9].

(2)Addressing limitations in internal and external resources through a three-pronged approach: a focus on patients, healthcare providers, and health systems and communities to improve access and the quality of care.

Patients with a history of trauma need multiple resources. Adverse experiences occur within the context of multiple socio-ecological interaction points, from the microsystem to the macrosystem level. Thus, TIC requires an interdisciplinary team of healthcare providers, e.g., professionals from other disciplines and sectors, researchers, policymakers, and the community [40]. Interdisciplinary collaboration is necessary, not only at the point-of-care and referral to specialists but also in linking patients to community resources and services.

(3)Practicing shared decision-making in trauma-informed care.

Shared decision-making equalizes the power differential between patients and physicians. Actively engaging individuals in the care process through a trusting and supportive partnership with their healthcare providers improves treatment adherence and increases patient satisfaction. The benefits of this therapeutic alliance are likewise observed among children and adolescents who have experienced trauma [73].

Trauma occurs within a unique context in the lives of individuals. Vital to shared decision-making is considering the patient’s voice and ethnic background to empower them to make concrete behavior changes. Thus, trauma’s multi-faceted nature calls for cultural competence on the part of physicians. Otherwise, symptoms of trauma may not only go unrecognized but also misinterpreted. This can result in ineffective interventions that do not match the needs of individuals from diverse racial and cultural backgrounds.

(4)Focusing on upstream family/household-centered interventions for building individual and collective resilience in coping with adversity.

Family routines and dynamics influence overall mental health and well-being. Despite the familial nature of mental health, existing treatment models all too often focus solely on the individual. Family-centered clinical approaches can help strengthen the family’s collective capacity to prevent adverse experiences even before they occur, address the factors that resulted in trauma, and strengthen individual and collective resilience [74,75].

(5)Addressing inequities in mental health issues to improve access to care among vulnerable and marginalized populations.

Physician–patient interaction presents an opportunity to uncover a history of trauma. It can be viewed as a natural and safe space to disclose such experiences and discuss related concerns [54]. However, individuals from vulnerable and marginalized communities experience inequities that limit their access to health services. Thus, not only are they less likely to see a physician for the care they need, but the trauma experienced by these populations can also go undiagnosed and unaddressed.

### 4.4. Limitations

This study offers the advantage of examining the insights of a national sample of U.S. physicians on TIC implementation in clinical settings. However, as an exploratory study, it also presents several limitations. The findings of this study should be considered based on these limitations.

First, the survey could have been enhanced by including additional demographic details such as the respondents’ duration of professional practice and the type of healthcare facility, including its urban–rural location.

Second, the small study size limits the generalizability of the findings of this study. The Reaction Data platform is typically used to distribute brief (pulse) surveys to healthcare professionals. As such, physician respondents in this sampling frame were used for shorter surveys. Other than email invitations and reminders, there were no other non-monetary or monetary incentives employed to increase survey participation.

Third, missing data (above 30% on some items), especially close to the end of the survey, may have been a factor in both the survey length and the lack of participation incentives. Together with physicians’ hectic work schedules, these factors may have disincentivized some respondents to complete the rest of the questions. Based on the study by Cunningham et al. [76], physician-specific web-based surveys typically have the lowest response rates that vary by specialty, with time constraints and survey burden being the primary deterrents to survey completion. The missing data and the lack of additional demographic information made it challenging to run further statistical analyses to identify specific demographic variables that may have a significant association with TIC implementation in clinical settings. Contextualizing physicians’ responses as to the geographic location of their respective facilities could have enriched the survey data. For example, responses to the question on the estimated percentage of patient caseload with a history of trauma, though subject to recall bias, could have been analyzed according to the type of healthcare facility and urban–rural location from the context of physician TIC training and resource allocation to at-risk populations. Zip code data could have also been asked, given that patient demographics vary by zip code. These details could be considered in the subsequent iteration of the survey. Despite these limitations, the small number of respondents who completed the survey may suggest limited TIC implementation in general. This may result from insufficiently addressed constraints that physicians face in their practice, such as inadequate TIC training, busy schedules, heavy patient caseload, paperwork completion, and staff shortages, including the overall lack of interest and/or incentives to practice TIC.

Fourth, while the survey included a demographic question on age, it unintentionally omitted the question on duration of practice. Age alone does not accurately correspond with the years of professional experience. Future studies could explore the relationship between practice experience, TIC receptivity, and medical specialty and how these variables may be influenced by physician burnout and personal trauma history. Additionally, the survey focused solely on physicians’ perspectives. It would be equally interesting to likewise obtain the viewpoints of patients and invite their assessments of the extent to which their respective physicians implement TIC.

Finally, the framing of the survey questions could be refined with consistent Likert scale options offered throughout the survey. The questionnaire was a faithful translation of the actual clinical strategies that comprised each of the five core TIC principles of the TIC pyramid [15]. However, this resulted in a variable number of survey questions per TIC principle, in which “Patient-Centered Communication and Care” had as many as six questions while “Understanding the Health Effects of Trauma” had only two questions. In addition, the TIC principle of “Understanding Your Own [Trauma] History and Reactions” had the lowest internal consistency, with a Cronbach’s alpha of 0.51, compared to the other TIC principles. This low inter-item correlation of the survey questions reflects the different TIC practices being examined under one principle: (1) reflecting on personal trauma history and how it impacts a provider’s interactions with patients; (2) recognizing when a provider’s emotional health is affected by a patient’s trauma history; (3) ways of caring for self and managing stress. The future iteration of the survey could focus on one of these concepts to reliably capture the attitudes of physicians on handling stress and personal trauma history.

## 5. Conclusions

Despite the widespread experience of trauma in the population, a trauma-informed approach has yet to be universally integrated into the clinical standard of care.

Applying a trauma-informed approach across different healthcare contexts and populations necessitates trained providers and staff who can effectively deliver such care. Data from this study may assist healthcare providers and administrators in identifying various strategies for enhancing physician TIC training and competence and in promoting cultural humility, which is vital to recognizing and understanding the cultural nuances of trauma. Patients, particularly from vulnerable and marginalized populations, can benefit from the enhanced TIC competence and sensitivity of their healthcare providers.

Concerted efforts at various organizational levels are necessary to seamlessly incorporate TIC within clinical practice guidelines. This study focused primarily on a facet of TIC implementation, i.e., examining the extent to which trauma-informed care practices were applied by physicians in clinical settings. However, true TIC involves a collective response at all levels of service and leadership. This requires a shared understanding of trauma and its impact not only on patients but also on providers and staff. It recognizes that trauma can likewise impact the capacity of its providers to offer TIC. This understanding emanates from the leadership echelon and permeates all layers of the organizational structure. In addition, TIC-committed leadership aligns its organizational policies with clinical practices. It provides regular system-wide TIC training, creates interdisciplinary trauma response teams, and promotes a platform for intersectoral dialogue on the implications of TIC on treatment delivery and health outcomes.

True TIC is marked by the adoption and delivery of a whole-of-care approach. With TIC as the mutual foundation and link to healthcare services, the aim is to provide a holistic framework for a physician–patient partnership that supports patient choice, trust, and empowerment while also fostering a TIC work culture of safety, emotional support, and resilience. This is not accomplished in a single setting nor through a single technique but through a multi-faceted and sustainable investment in a trauma-informed workforce and leadership. In doing so, TIC has the potential to increase the quality of care, reshape the delivery of care, and ultimately offer affordable and culturally sensitive care [15,58].

## Figures and Tables

**Figure 1 ijerph-21-00232-f001:**
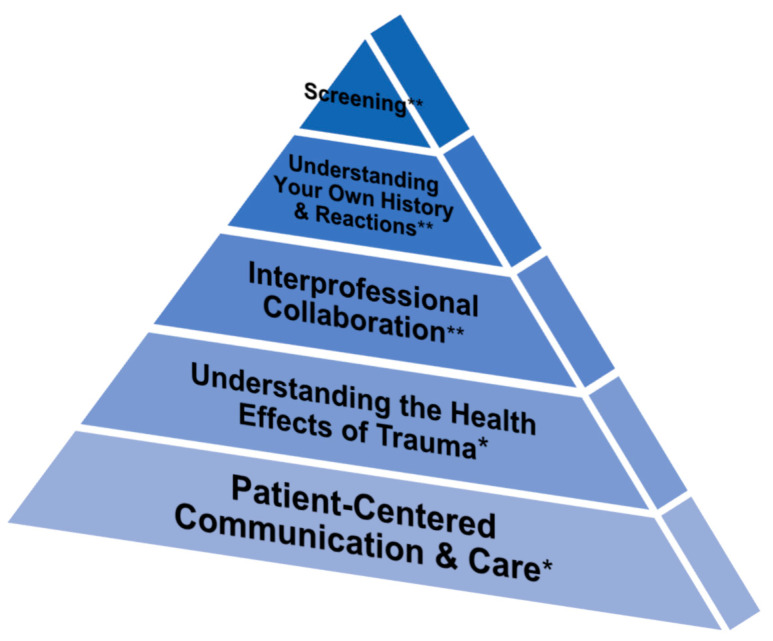
Trauma-informed care (TIC) pyramid, Raja et al. [15]. * Universal trauma precautions domain and ** trauma-specific care domain.

**Table 1 ijerph-21-00232-t001:** Demographic characteristics of the study population.

Demographic Characteristics	N = 179 n	%
**Age**, n = 112		
36–45	4	3.6
46–55	25	22.3
56–65	48	42.9
66–75	24	21.4
≥75	9	8.0
Preferred not to answer	2	1.8
**Race**, n = 98		
Asian or Pacific Islander	9	9.2
Black or African American	2	2.0
Middle Eastern	1	1.0
Multiracial or Biracial	4	4.1
White or Caucasian	82	83.7
**Gender**, n = 112		
Male	75	67.0
Female	35	31.3
Transgender Female	1	0.9
Preferred not to answer	1	0.9
**Employment Position**, n = 179		
Primary Care Physician (General Internal Medicine, Pediatrics, Obstetrics/Gynecology, Complementary Medicine (Chiropractic Care))	80	44.7
Specialty Physician (Surgery and Anesthesiology, Critical Care Medicine, Reproductive Medicine, Ophthalmology, Otolaryngology, Psychiatry/Neurology, Cardiology, Dermatology, Gastroenterology, Hematology, Oncology, Nephrology/Urology)	92	51.4
Medical Director/Chair	7	3.9

**Table 2 ijerph-21-00232-t002:** Perceived barriers to TIC implementation in clinical settings.

Perceived Barriers to TIC Implementation	Frequency n	(%)
1. **Resource Constraints**, N = 117 *(Limited in-network services, long wait lists for services, lack of culturally and linguistically appropriate services, and lack of geographically accessible services and resources)*	62	53.0
2. **Referral Limitations/Interdisciplinary Collaboration**, N = 117 *(Lack of knowledge on the right provider to refer patients with a traumatic history)*	55	47.0
3. **Time Constraints**, N = 118 *(Scheduling conflicts, short appointment time, high patient load, and paperwork)*	55	46.6
4. **Administrative Constraints**, N = 117 *(Expectations for seeing a high patient load or completing paperwork)*	49	41.9
5. **Difficulty Managing Personal Stress and/or Practicing Self-Care**, N = 117	45	38.5
6. **Inadequate TIC Emphasis in Medical Education/Training**, N = 118 *(Lack of emphasis on TIC during medical education/training)*	45	38.1
7. **Lack of Institutional Support**, N = 118 *(No clear institutional process to establish TIC)*	42	35.6
8. **Slow Implementation of Change**, N = 118	38	32.2
9. **Personal Trauma History**, N = 116 *(Lack of access to mental health support or feeling activated/triggered by trauma-related topics)*	31	26.7

**Table 3 ijerph-21-00232-t003:** Extent of physician training on the core principles of trauma-informed care (TIC) in healthcare based on Raja et al.’s TIC pyramid (multiple-response option).

Training on the TIC Pyramid * Core Principles in Healthcare	N = 112 n (%)
Training in 1 TIC Core Principle	17 (15.2)
Training in 2 TIC Core Principles	19 (17.0)
Training in 3 TIC Core Principles	27 (24.1)
Training in 4 TIC Core Principles	14 (12.5)
Training in 5 TIC Core Principles	26 (23.2)
Preferred not to answer/Skipped question	9 (8.0)

* Raja et al.’s [15] core TIC principles of (1) patient-centered communication and care, (2) understanding the health effects of trauma, (3) interprofessional collaboration, (4) understanding your own history and reactions, and (5) screening.

**Table 4 ijerph-21-00232-t004:** Most frequently reported training on the core principles of trauma-informed care (TIC) in healthcare based on Raja et al.’s TIC pyramid (multiple-response option).

TIC Pyramid * Core Principles in Healthcare	Training Frequency N = 324 ** n (%)
Universal Trauma Precautions	
Patient-Centered Communication and Care	83 (25.6)
Understanding the Health Effects of Trauma	74 (22.8)
Trauma-Specific Care	
Screening (for traumatic events)	74 (22.8)
Interprofessional Collaboration	62 (19.1)
Understanding Your Own History and Reactions	31 (9.6)

* Raja et al., 2015; ** Based on the total number of responses received to a multiple-response question in which respondents selected several responses as applicable.

**Table 5 ijerph-21-00232-t005:** Practicing TIC patient-centered communication and care.

Patient-Centered Communication and Care (Cronbach’s Alpha = 0.75)		Likert Scale				
	Mean Score (SD) ****	Always n ** (%) 5 ***	Often n (%) 4	Sometimes n (%) 3	Rarely n (%) 2	Never n (%) 1
*I ask what can be done to make patients more comfortable during the appointment* (N * = 175)	2.5 (1.3)	52 (29.7)	45 (25.7)	38 (21.7)	24 (13.7)	16 (9.1)
*Prior to a physical examination, I present a brief summary of what parts of the body will be examined and allow the patient to ask questions* (N = 171)	2.2 (1.4)	85 (49.7)	28 (16.4)	23 (13.5)	15 (8.8)	20 (11.7)
*I give the option of shifting an item of clothing out of the way rather than putting on a gown when an entire area does not need to be examined* (N = 172)	2.1 (1.4)	89 (51.7)	35 (20.4)	19 (11.1)	8 (4.7)	21 (12.2)
*I offer patients a pillow for their back if they are anxious about being in the supine position* (N = 172)	2.4 (1.5)	73 (42.4)	28 (16.3)	24 (14.0)	16 (9.3)	31 (18.0)
*I give patients the option of a mirror for patients to see procedures or examinations that are out of their field of vision* (N = 172)	3.7 (1.6)	30 (17.4)	15 (8.7)	20 (11.6)	23 (13.4)	84 (48.8)
*I offer patients a way to “signal” anxiety verbally or nonverbally (e.g., raising their hand) during procedures/exams (e.g., Pap smear)* (N = 171)	2.7 (1.6)	57 (33.3)	31 (18.1)	28 (16.4)	18 (10.5)	37 (21.6)

* N = Total number of participants who responded to the survey question; ** n = number of participants who selected a particular Likert scale option; *** = sentiment score; **** SD = standard deviation.

**Table 6 ijerph-21-00232-t006:** Provider understanding of the health effects of trauma.

Understanding the Health Effects of Trauma (Cronbach’s Alpha = 0.79)		Likert Scale				
	Mean Score (SD) ****	Always n ** (%) 5 ***	Often n (%) 4	Sometimes n (%) 3	Rarely n (%) 2	Never n (%) 1
*I discuss with patients the relationship between unhealthy behaviors (maladaptive coping methods) and stress/trauma* (N * = 159)	2.8 (1.2)	29 (18.2)	40 (25.2)	49 (30.8)	23 (14.5)	18 (11.3)
*When discussing behaviors that are detrimental to the patient’s wellbeing, I brainstorm with them potential solutions for how to change the behavior* (N = 158)	2.6 (1.2)	32 (20.3)	47 (29.8)	39 (24.7)	25 (15.8)	15 (9.5)

* N = Total number of participants who responded to the survey question; ** n = number of participants who selected a particular Likert scale option; *** = sentiment score; **** SD = standard deviation.

**Table 7 ijerph-21-00232-t007:** Practicing interprofessional collaboration on trauma-informed care.

Interprofessional Collaboration (Cronbach’s Alpha = 0.71)		Likert Scale				
	Mean Score (SD) ****	Strongly Agree n ** (%) 5 ***	Agree n (%) 4	Neither Agree nor Disagree n (%) 3	Disagree n (%) 2	Strongly Disagree n (%) 1
*I maintain a list of referral sources for patients who disclosed a trauma history* (N * = 150)	2.6 (1.5)	48 (32.0)	36 (24.0)	21 (14.0)	19 (12.7)	26 (17.3)
*I am confident in sensitively referring a patient with trauma* (N = 151)	2.4 (1.3)	51 (33.8)	39 (25.8)	28 (18.5)	16 (10.6)	17 (11.3)
*Referral and educational materials about trauma are readily available in the waiting room for patients* (N = 150)	3.2 (1.6)	32 (21.3)	20 (13.3)	20 (13.3)	29 (19.3)	49 (32.7)
*I am confident in working with nurses, medical interpreters, first responders, and others when caring for patients who have experienced trauma* (N = 152)	2.4 (1.4)	54 (35.5)	30 (19.7)	38 (25.0)	9 (5.9)	21 (13.8)

* N = Total number of participants who responded to the survey question; ** n = number of participants who selected a particular Likert scale option; *** = sentiment score; **** SD = standard deviation.

**Table 8 ijerph-21-00232-t008:** Physicians’ understanding of their personal trauma history and reactions.

Physicians’ Understanding of Their Personal Trauma History and Reactions (Cronbach’s Alpha = 0.51)		Likert Scale				
	Mean Score (SD) ****	Strongly Agree n ** (%) 5 ***	Agree n (%) 4	Neither Agree nor Disagree n (%) 3	Disagree n (%) 2	Strongly Disagree n (%) 1
*I reflect on my own stress and/or trauma history and how it may influence patient interactions* (N * = 120)	2.4 (1.1)	27 (22.5)	47 (39.2)	28 (23.3)	12 (10.0)	6 (5.0)
*I recognize when caring for patients with a trauma history begins to impact my own emotional health and wellbeing* (N = 120)	2.2 (0.9)	26 (21.7)	60 (50.0)	22 (18.3)	9 (7.5)	3 (2.5)
		Always n ** (%) 5 ***	Often n (%) 4	Sometimes n (%) 3	Rarely n (%) 2	Never n (%) 1
*I practice self-care strategies (e.g., exercise, social support, etc.)* (N = 120)	2.0 (1.0)	42 (35.0)	43 (35.8)	28 (23.3)	4 (3.3)	3 (2.5)
*I use counseling/mental health services to help me manage stress when I experience high levels of stress* (N = 119)	3.6 (1.3)	10 (8.4)	15 (12.6)	25 (21.0)	31 (26.1)	38 (31.9)

* N = Total number of participants who responded to the survey question; ** n = number of participants who selected a particular Likert scale option; *** = sentiment score; **** SD = standard deviation.

**Table 9 ijerph-21-00232-t009:** Screening for trauma in new patients and in patients with functional challenges.

Screening (Cronbach’s Alpha, 5 items = 0.79)		Likert Scale				
	Mean Score (SD) ****	Strongly Agree n ** (%) 5 ***	Agree n (%) 4	Neither Agree nor Disagree n (%) 3	Disagree n (%) 2	Strongly Disagree n (%) 1
*It is important to assess every case of trauma* (N * = 142)	2.6 (1.3)	69 (48.6)	47 (33.1)	19 (13.4)	4 (2.8)	3 (2.1)
		Always n ** (%) 5 ***	Often n (%) 4	Sometimes n (%) 3	Rarely n (%) 2	Never n (%) 1
*I screen for trauma in every new patient* (N * = 144)	1.8 (0.9)	36 (25.0)	35 (24.3)	33 (22.9)	28 (19.4)	12 (8.3)
*I screen for current trauma or a history of traumatic events as a follow-up to the universal screening questions* (N = 143)	2.6 (1.3)	34 (23.8)	38 (26.6)	35(24.5)	19 (13.3)	17 (11.9)
*I (or my office staff) prepare a patient for potentially difficult questions prior to a trauma screening (e.g., reviewing confidentiality)* (N = 144)	2.7 (1.4)	38 (26.4)	30 (20.8)	34 (23.6)	22 (15.3)	20 (13.9)
*Communication skills training is provided to all office staff about how to sensitively talk to patients who disclose a history of trauma* (N = 143)	2.8 (1.3)	33 (23.1)	32 (22.4)	33 (23.1)	27 (18.9)	18 (12.6)

* N = Total number of participants who responded to the survey question; ** n = number of participants who selected a particular Likert scale option; *** = sentiment score; **** SD = standard deviation.

## Data Availability

Data were obtained from Reaction Data and are available from M.L.B.N. and C.L.H. with the permission of Reaction Data.

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
