# Peer review of "U.S. Physicians’ Training and Experience in Providing Trauma-Informed Care in Clinical Settings"

_ijerph, 2024, doi:10.3390/ijerph21020232_

Round 1

Reviewer 1 Report

Comments and Suggestions for Authors

The authors present a cross-sectional survey of trauma-informed care training needs with physicians. The paper requires major revision to tighten up the design and findings. 

The paper will be of interest and could be worthwhile publishing if some of the following is considered and incorporated into the article.  I have various concerns which I will bullet point, many of these concerns have to do with the use of different and interchangeable terms to describe trauma, and the validity of what physicians are subjectively assessing.

1.0 introduction 

The authors seem to be using trauma and adversity interchangeably. It's important to distinguish between both concepts and be very clear about this. Might be an idea to define the terms being used.

P.2 Again, is it trauma or adversity, the terms are used interchangeably 

1.2 Might be an idea to define trauma earlier on in the article

1.2 Bringing PTSD diagnosis into it complicates things further, the authors needed to be clear on what their article is about and how the terms used relate to the study. Trauma et al is fast becoming everything and anything, it is incumbent on researchers to define and be clear about this, especially as it relates to the study which I address further in the article

1.2 P.3 Line 102

This sounds like complex trauma, however, it is not explicit.

1.4 through to 1.4.3 can be summarised  in less words, it runs on a bit

3.2.1 

The event is not traumatic, trauma is the experience. There needs to be more precision throughout the manuscript, and this is also an issue in how the sample are assessing 'trauma'.

3.2.1

This is quite a big limitation of the study. Respondents are asked to 1) estimate prevalence, 2) they are not basing this on screening or use of a diagnostic instrument and as such, cannot speak to the prevalence of trauma, but only of events. I would like to see much more on this in the limitation section and also reference to it in the literature review.

table 2

Trauma-informed leadership is essential for implementation, surprising not to see it in the pyramid, what are the limitations for implementation?

table 8

Cronbach's of 0.51 is very low internal consistency, needs to be acknowledged as a limitation of the questionnaire

4.1 P.14

Secondary traumatic stress does not affect non-clinical staff, it is specific to clinical staff who repeatedly are exposed to trauma content. Also, the references used to support this statement do no support  this or use primary data

4.5 P18

The framework of TIC used to analyse findings in this study may not have been the model that participants were trained in, and as such presents limitations.

Comments on the Quality of English Language

N/A

Author Response

Dear Editors,
         We would like to thank Reviewer 1 and Reviewer 2 for their time, expertise, and comments. We recognize that there is room for improvement in our paper. We have used their careful reviews, borne by both expertise and experience, to guide our revisions. By addressing the suggestions of both reviewers, it provided greater clarity to the articulation of our findings.

         Please see below our point-by-point response to the reviewers. We believe that we have sufficiently addressed their concerns and comments. We hope that our careful review and revisions will further strengthen our paper and meet the high standards of publication of MDPI’s IJERPH.

Reviewer 2 Report

Comments and Suggestions for Authors

Thank you for sharing an interesting and valuable manuscript. I found the manuscript well-written and comprehensive. I just have a few comments for consideration:

1. On page 2, line 96, where you mention that not all who experience trauma develop PTSD. Might you add a sentence or two about factors that may prevent one person from developing PTSD compared to another person who experienced the same event experiencing PTSD?

2. On page 3, I would suggest you add a figure of the TIC pyramid. Sometimes visuals are helpful aids to communicate concepts.

3. On page 4, line 152, I would suggest your adding "and avoiding or altering" after "recognizing." Believe good clinical TIC would allow for some altering of procedures that may be highly traumatizing?

4. On page 4, line 156, I would add the word "Clinical" at the beginning of the sentence. Believe your manuscript focuses on clinical vs organizational trauma-informed care.

5. On page 5, line 222, was a random national sample used? If so, please indicate so. 

6. On page 5, line 233, what is the sampling frame used by Reaction Data? It is important to be very specific about the sampling frame and what types of study participants are in their database.

7. On page 6, line 266, how many physicians were invited to participate in the study? And what was the response rate (i.e., percentage) that participated? Any information about how well the participants reflected the demographics of the total sample population would also be helpful to note (was there some selection bias with those who participated compared to all potential respondents?).  

8. In Table 5 (through 9), I see you have a footnote about sentiment value. I am not familiar with this term and am wondering if another term might be used to reflect this concept?

9. On page 16, line 592 to 593, you discuss the item "Understanding Own [Personal Trauma] History and Reactions." The items in this subsection of the survey reflect different concepts - contributing to a low Cronbach's alpha. It is both about how physicians reflect on their own trauma histories and how interaction with some patients may trigger a potentially negative response based on the physician's own trauma history, as well as self-care and managing stress. I believe in the next iteration of this survey that it would be helpful to separate these two types of questions, and by so doing, I am sure the Cronbach's alpha will improve. A more immediate suggestion is to discuss how a physician's trauma history may affect their ability to provide TIC and what some vulnerabilities might exist for a physician with a significant trauma history (as well as supports that can be put in place).  

10. Lastly, believe your manuscript spoke mostly about TIC from a clinical perspective. It would be helpful if you can add a few sentences at the end of the manuscript about how true TIC in a setting is holistic - meaning that the organization itself needs to be a trauma-responsive organization in which physicians and staff are treated in a trauma-informed way, so that they are best able to provide TIC to their patients. There is a parallel process, in that staff need to first feel like they are in a trauma-responsive setting, which will provide a supportive context in which they can provide best TIC clinical services. Hope this makes sense and perhaps making this distinction in the Limitations section will be valuable to the reader (and a more accurate reflection of a true TIC system).

Thank you again for your valuable work and sharing your manuscript. I hope my comments are somewhat helpful.  

Author Response

(The authors gave the same response as above.)

Round 2

Reviewer 1 Report

Comments and Suggestions for Authors

The authors have largely made the changes suggested